# Examination of Multi-Pulse Rectifiers of PES Systems Used on Airplanes Compliant with the Concept of Electrified Aircraft

Lucjan Setlak and Rafał Kowalik *

Faculty of Aviation Division, Department of Avionics and Control Systems, Polish Air Force University, 08-521 Deblin, Poland; l.setlak@law.mil.pl
* Correspondence: r.kowalik@law.mil.pl

**Abstract:** This article focuses on power electronic multi-pulse 12-, 24- and 36-impulse rectifiers based on multi-winding rectifier transformers. The effectiveness of voltage processing with different variants of supply voltage sources is discussed and arguments are formulated for limiting oneself to 24-pulse processing, which is used in the latest technological solutions of modern aviation technology. The main purpose of this article is to conduct a study (analysis, mathematical models, simulations) of selected multi-pulse rectifiers in the context of testing their properties in relation to the impact on the electrified power supply network. The secondary objective of the article is to assess the possibility of using Matlab/Simulink to analyze the work of rectifier circuits implemented in aircraft networks compliant with the more/all electric aircraft (MEA/AEA) concept. The simulation tests included designing a typical auto-transformer rectifier unit (ATRU) system in the Simulink program and generating output voltage waveforms in this program in the absence of damage to the rectifier elements. In the final part of this work, based on a critical analysis of the literature on the subject of the study, simulations were made of exemplary rectifiers in the Matlab/Simulink programming environment along with their brief analysis. Practical conclusions resulting from the implementation of the MEA/AEA concept in modern aviation were formulated.

**Keywords:** examination; multi-pulse rectifiers; energo-electronic power system (PES); more/all electric aircraft

## 1. Introduction

Currently, in modern aviation, military (Lockheed Martin) and civil aviation (Airbus, Boeing), both in the field of military aircrafts JSF (Joint Strike Fighter) F-35 and F-22 Raptor, as well as civilian aircrafts (A-380 and A-350 XWB, B-787), according to the evolving trend of the more/all electric MEA/AEA aircraft (More/All Electric Aircraft), the dynamic development of on-board advanced electrical systems AAES (Aircraft Advanced Electrical Systems) can be observed. The above systems are characterized by advanced technological solutions in terms of two key components in the form of systems, namely the electro-energetic power supply system EPS (Electric Power System) and energo-electronic power system PES (Power Electronics Systems) and its main components, such as multi-pulse rectifiers, including 12-, 24- and 36-pulse, which were subject to detailed analysis in this paper. A representative of multi-pulse rectifiers in aviation is a transformer-rectifying device, popularly referred to as a TRU (Transformer Rectifiers Unit), used primarily in the absence of basic DC (Direct Current) sources (generator, integrated starter/generator unit) and ATRU (Auto-Transformer Rectifier Unit) on-board aircrafts.

They are mainly used on aircrafts where the leading power system is a constant-frequency AC (Alternating Current) electro-energetic power supply system in the case of conventional aircrafts, and

an on-board autonomous power system for alternating frequency AC in the case of advanced aircrafts compatible with the MEA/AEA concept. Changes in the on-board power supply in modern aviation exert a decisive influence on the dynamically developing concept of a more electric aircraft, including in particular the dominant role of innovative systems in the generation, processing and transmission of AC electricity on-board a modern aircraft. Referring to the analysis of the dominant role of an on-board AC power supply, it should be noted that in the case of advanced aircrafts in accordance with the MEA/AEA concept, the electrical network is based on VF (Variable Frequency) or integrated AC unit starter/alternator AS/G VF (Alternating Starter/Generator Variable Frequency).

In the case of aircrafts in the field of "More Electric" or MET (More Electric Technology), advanced AC electric networks of elevated frequency in the range of 360-800 Hz are used in various types of industries (marine, automotive, renewable energy sources) and above all in aviation and space applications in which the key essence consists of converting conventional types of energy (electric, mechanical, hydraulic, pneumatic) to one kind of energy, namely electricity. Usually, especially in the case of classical aircrafts, the systems considered above are supplemented with DC electric networks, using voltages of different values, suited to the target receivers working on-board aircrafts [1,2].

The current state of the technology, especially of electronics and power electronics, allows for the reduction of a number of power systems to one system, namely to AC mains, leaving emergency sources based on accumulator batteries as the so-called final link of the on-board power supply of the modern aircraft. Through modern advanced systems such as electronic devices and power electronic elements, it is possible to design rectifier circuits matching one AC network to provide power to all receivers, regardless of the value and type of voltage (AC or DC). However, the problem with creating a uniform power system and the use of semiconductor converters requires a number of tests to obtain the appropriate PF (Power Factor), determined by the quotient of active power to maximum power found in the power supply system and the THD (Total Harmonic Distortion) signal aspect ratio. The power factor is an element of economic effect, and the essence of the shape coefficient of signals is the maintenance of electromagnetic compatibility. In a further part of this article, a review and preliminary analysis of multi-pulse rectifiers used in TRU and ATRU units is presented [3,4].

Based on the above, it can be concluded that the problem of multi-pulse rectifiers is very important in the area of the ever-increasing trend of the electrified aircraft. The concept of a more electric aircraft has been known for a long time, but only with the development of power electronic devices, components and systems based on advanced semiconductor technology, has it been implemented to modern aviation in the field of advanced power systems. Systems of this type use variable frequency sources as well as high voltage direct current (HVDC) architecture and alternating current HVAC (High Voltage Alternating Current).

In turn, the implementation of the trend of a more electric aircraft contributed, among others, to increase the efficiency of the entire aircraft in terms of increasing its reliability, flexibility and economy. Technological solutions in line with the MEA/AEA concept are constantly evaluated and have enormous potential in terms of weight, quality and operating costs.

## 2. Review and Analysis of Multi-Pulse Rectifiers Used in the TRU and ATRU Units in the Field of the PES System in Accordance with the MEA Concept

Currently, in technologically advanced aviation (military, civilian), and increasingly in designed on-board autonomous power systems, one can observe a dynamic trend in the development of power electronic multi-pulse rectifiers in the PES system, particularly on aircrafts compatible with the concept of the electrified aircraft. Multi-pulse rectifiers (6-, 12- and 18-, 24-impulse), determined for the needs of, among others, aviation issues as so-called transformer-rectifier devices TRU, are a key component of the power electronic system of contemporary military aircrafts (F-22 Raptor and F-35), as well as civilian ones (A-380 and A-350 XWB, B-787), and play a key role in the implementation of the trend of a more electric aircraft.

When performing a preliminary analysis of rectifiers used by TRU and ATRU units, it should be noted that aviation systems converting AC to DC current (AC to DC) used in aviation are obliged to provide "rectified" voltage in two different ranges in the electricity network of more electric aircraft (MEA). They consist of multiphase generators and rectifier circuits. Rectifiers of this type are power electronics systems designed to convert AC voltages into DC (unidirectional) voltages. Usually they are powered by single or three-phase sinusoidal voltages. With reference to the time waveform of the unidirectional output voltage, determined by the rectified voltage whose value depends on the type of rectifier (voltage drops), the properties of the applied filter and the load affecting its efficiency consist of appropriate sine voltage sections supplying the rectifier.

DC voltages received from rectifier circuits always contain higher harmonics, which negatively affect the operation of drive systems of electric engines. The systems responsible for converting AC voltage to DC are usually non-linear systems that convert electrical energy and cause distorted currents and voltages in the electrical network of the aircraft. In general, AC distribution systems require a longer processing time than DC systems.

It should be noted that the conversion sequence is short (or even non-existent) if the AC sources and loads are connected to the AC mains, and the DC sources and loads are connected to the DC network. In such cases, processing equipment is unnecessary and there are no associated costs. In turn, higher harmonics of substation load current cause additional voltage drops in the substation power supply network and the resultant voltage is always deformed. Hence, there is a need to minimize the harmonic content in the power supply in ATRU systems.

For example, if there are six-pulse bridge systems in the ATRU unit, they are responsible for generating harmonic signals on the network of 5, 7, 11 and 13 with a level of up to 20% of the input voltage basic component. In view of the above, for the limitation of harmonics, it is possible to use, for example, three-phase AC current chokes on the side of the supply voltage. The choke increases the reactance of the line supplying the electronic apparatus of the aircraft for all harmonics, affects the times of control circuits in the rectifier systems, and also reduces the voltage of the intermediate circuit. The load current of the substation is influenced by the recipient through the use of systems with a larger number of pulses; thus, by reducing the pulsation of "rectified" voltage, the negative impact on the mains voltage should be limited. For this purpose, multi-pulse rectifier systems are used, e.g. 18- and 24-pulse systems. In AC systems with distributed generation, the generated DC current must be converted using DC/AC converters.

In addition, it should be added that if the generated electricity is consumed by AC loads, then further conversion is unnecessary. For DC end-of-flight aircraft, the flow map includes DC/AC conversion and a suitable transducer is required. For DC loading, the sequence is DC/AC/DC and both transducer and rectifier are required. In turn for AC loads with a converter, the sequence DC/AC/DC/AC is valid and a sequence is required: DC/AC converter, AC/DC converter and DC/AC converter [5–7].

## 2.1. Twelve-Impulse Rectifiers

Schematic diagram of a typical 12-pulse rectifier in a parallel system on the example of an ATRU unit is shown in the figure below (Figure 1). It should be noted that the independent operation of the units in the parallel connection is ensured by matching chokes. The alternating current component of individual systems does not exceed 4%. The value of this current results from the value of the magnetizing current [8–10].

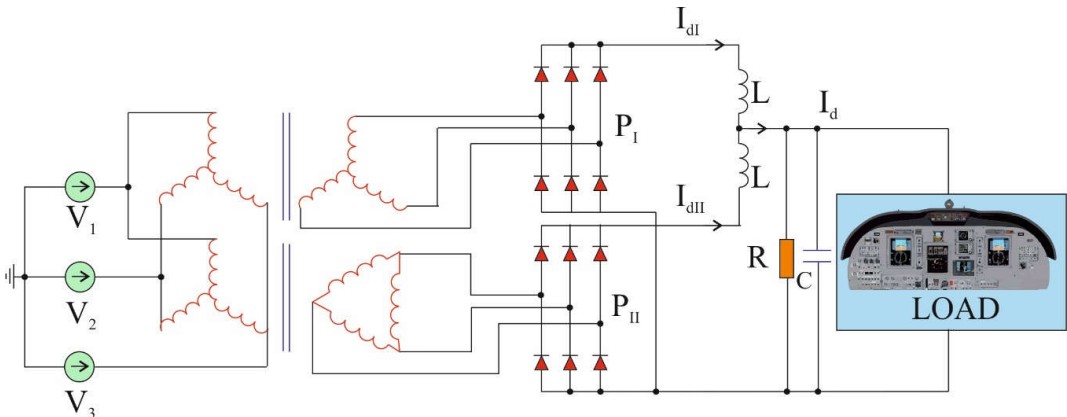

**Figure 1.** Diagram of a classical 12-pulse rectifier in the auto-transformer rectifier unit (ATRU) system.

Where: $P_1$ and $P_2$- ATRU rectifier circuits, each consisting of six diodes, $V_1$, $V_2$ and $V_3$- three-phase voltage sources, $I_{dI}$, $I_{dII}$- currents at the output of ATRU rectifier circuits, L, R and C- inductance, resitance and capacitance, appearing on terminal elements, $I_d$- current on terminal devices, LOAD- terminal devices.

### 2.2. Twenty Four-Impulse Rectifiers

The next figure (Figure 2) presents a modified system with a parallel connection of six-pulse rectifiers, exhibiting the features of a 24-pulse rectifier in the ATRU system of a more electric aircraft [11–13].

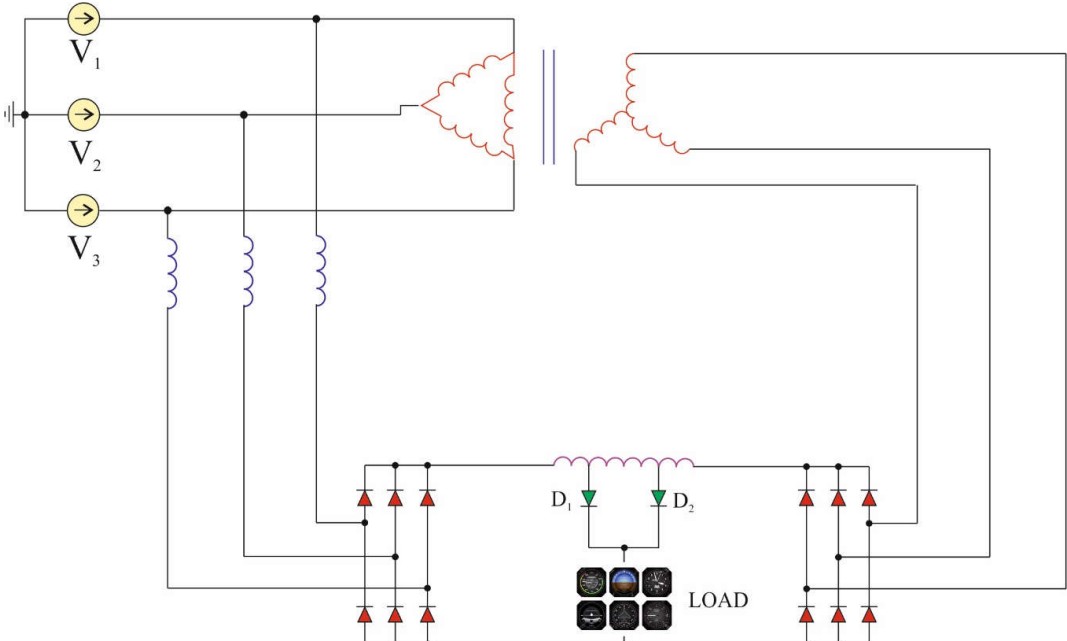

**Figure 2.** Diagram of a classic 24-pulse rectifier powered by a three-phase transformer.

Where: $D_1$ and $D_2$- ATRU rectifier diodes, $V_1$, $V_2$ and $V_3$- three-phase voltage sources, LOAD-terminal devices.

### 2.3. Thirty Six-Impulse Rectifiers

The system with a parallel connection of two six-pulse rectifiers showing the properties of the 36-pulse system is shown in the figure below (Figure 3). This is achieved by switching on three

thyristors in the order: T3-T1-T2-T2-T1-T3-T3-T1. The sequence of switching thyristors on in a specific order plays an important role due to the production of appropriate voltage and current components reaching the final equipment (devices, installations, systems) mounted on-board modern aircrafts. This condition must also be met due to the elimination of higher harmonics that negatively affect the AC voltage of the electrified aircraft's on-board network.

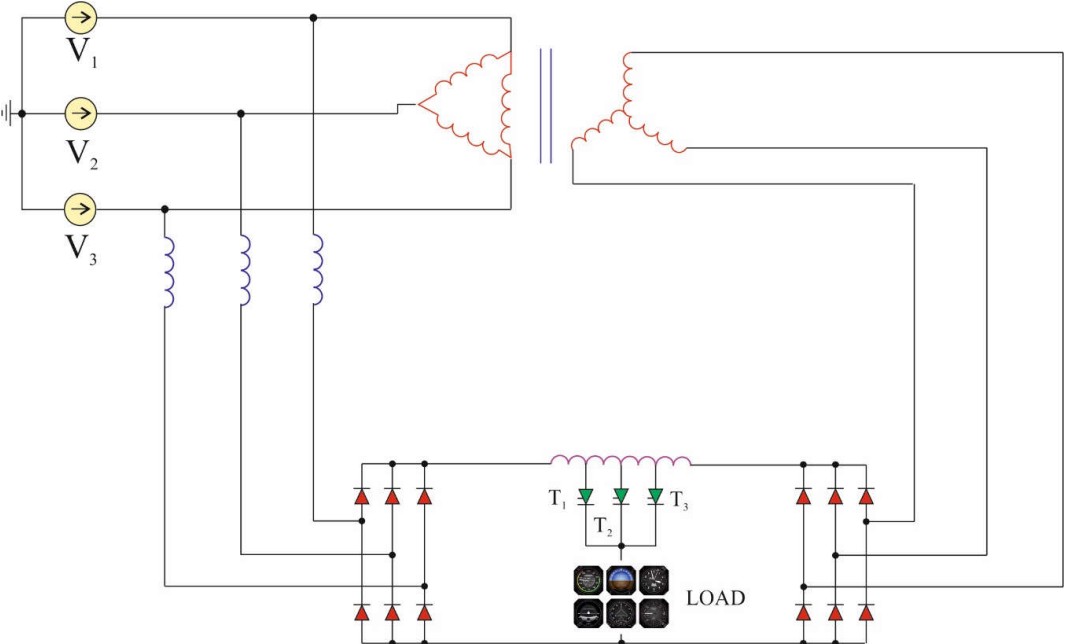

**Figure 3.** Diagram of a classic 36-pulse rectifier powered by a three-phase transformer.

Where: $T_1$, $T_2$ and $T_3$- thyristors, $V_1$, $V_2$ and $V_3$- three-phase voltage sources, LOAD- terminal devices.

The proper sequence of switching on thyristors also results in better power efficiency in the field of its transfer to terminal equipment. In addition, the divided windings of the matching choke in the DC circuit provide natural commutation of the attached thyristors [14,15].

## 3. Mathematical Model for Controlling the ATRU Unit in the Field of the PES System in Accordance with the Concept of MEA

Currently, the most commonly used technique responsible for the process of controlling the work of circuits (used to convert AC three-phase voltages and currents to their DC equivalents) is realized by PWM (Pulse Width Modulation) and rectifier circuits. Its main advantages, which influenced the use of this method in more electric aircrafts were, among others: the possibility of stabilizing the input voltage, elimination of undesirable harmonic signals in the output voltage and a more effective PF. All the above-mentioned factors have a decisive influence on the overall functioning of the electrical network of the more electric aircraft, in particular in the field of improving the quality of the electro-energetic system.

Subsequently, a mathematical model was created, describing the work of the PWM pulse width rectification circuit. During the process of creating a mathematical model of the rectifier, its key parameters (voltage, current) should first be considered. By analyzing these parameters in the scope of the numerical analysis process, it can be used to describe the mathematical method of converting AC voltage and current into their equivalents in the form of DC current.

The whole process was analyzed on the basis of the more electric aircraft on-board network, based on the ATRU subunit. The input voltages supplied to the ATRU system were delivered from an alternating current AC generator. The key goal of the mathematical analysis was to determine the stabilized DC output voltage in a short response time. The entire analysis process refers to the change in

the coordinate system, which determines the position of the PMSM (Permanent Magnet Synchronous Machine) poles *d-q* [16,17]. The diagram on the basis of which a description of mathematical electrical phenomena taking place in the ATRU system is shown in the figure below (Figure 4).

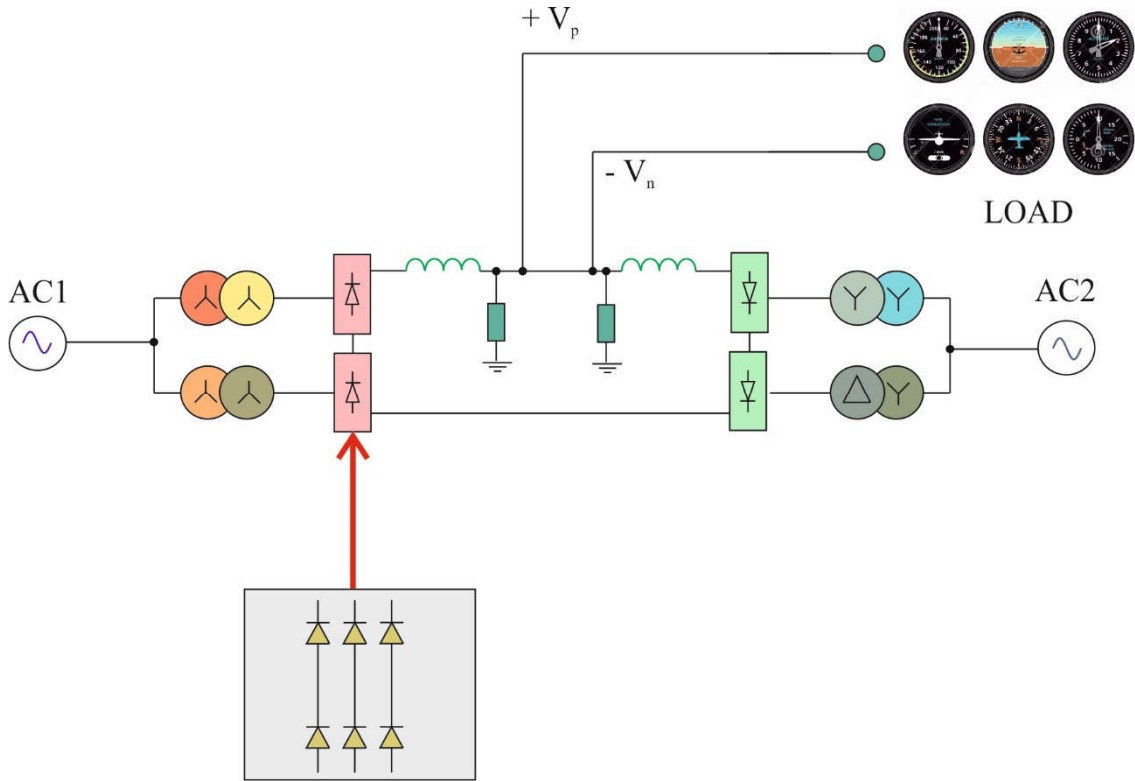

**Figure 4.** Diagram of a classic 36-pulse rectifier powered by a three-phase transformer.

Where: AC1 and AC2- three-phase sources AC (generators), $V_p$ and $V_n$- voltage value at the output of the system, LOAD- terminal devices.

Turning to the mathematical description, it should take into account the three-phase voltage given to the input of the ATRU converter in the reference system *d-q*. The following expression will take the form [18,19]:

$$
\begin{aligned}
u_{SA}(t) &= U_m \cos(\omega t) \\
u_{SB}(t) &= U_m \cos\left(\omega t - \tfrac{2}{3}\pi\right) \\
u_{SC}(t) &= U_m \cos\left(\omega t + \tfrac{2}{3}\pi\right)
\end{aligned}
\tag{1}
$$

where: $U_m$—is the amplitude of the maximum three-phase voltage of the generator AC, $u_{SA}(t)$, $u_{SB}(t)$ and $u_{SC}(t)$—are the control signal voltages in individual phases given to the converter input ATRU, and $\theta = \omega t$, $\alpha$- is the phase angle of the initial variable signal.

The mathematical record of the equations showing the induced voltages in each phase and the capacitive current can be written as follows:

$$
\begin{aligned}
L\frac{di_{SA}}{dt} &= u_{SA} - f_A U_{DC} \\
L\frac{di_{SB}}{dt} &= u_{SB} - f_B U_{DC} \\
L\frac{di_{SC}}{dt} &= u_{SC} - f_C U_{DC} \\
C\frac{du_{DC}}{dt} &= f_A i_{SA} + f_B i_{SB} + f_C i_{SC} - I_{LOAD}
\end{aligned}
\tag{2}
$$

where: L and C- are the inductance and capacity, occurring in the rectifier system, $f_A, f_B, f_C$- represent the switching function of the AC/DC converter, and $i_{SA}(t)$, $i_{SB}(t)$ and $i_{SC}(t)$- define the control signal

currents in a particular phase, respectively, given to the input of the ATRU converter, while $I_{LOAD}$ and $U_{DC}$- indicate the values of the load current and the DC voltage at the output of the rectifier.

The mathematical dependencies describing the operation of the function are as follows:

$$
\begin{aligned}
f_A &= \frac{(2S_A - S_B - S_C)}{3} \\
f_B &= \frac{(2S_B - S_A - S_C)}{3} \\
f_C &= \frac{(2S_C - S_A - S_B)}{3}
\end{aligned}
\tag{3}
$$

Parameters $S_A$, $S_B$, $S_C$- represent a control signal that is used to control the ATRU system in any phase of the input voltage. Each of the control signals emits a "0" or "1" state, which means if in the control system marked by $S_A$ there is a value of "1", the upper switch operates in phase A and is "switched on" (bottom switch in voltage phase A is "off"). Otherwise, when $S_A$ is set to state "0", the upper switch in phase A is "off" and the lower switch is "on".

The process of switching modes of operation in the ATRU system of the aircraft electrical network in accordance with the MEA concept is mainly based on the value of three-phase voltage; in particular, its phase. In turn, the vector defining the operating state of the generator is transformed in the two axes of the reference system *d-q*. The notation is as follows:

$$
\begin{bmatrix} X_d \\ X_q \end{bmatrix} = \frac{2}{3} \cdot \begin{bmatrix} \cos \omega t & \cos\left(\omega t - \frac{2}{3}\pi\right) & \cos\left(\omega t + \frac{2}{3}\pi\right) \\ -\sin \omega t & \sin\left(\omega t - \frac{2}{3}\pi\right) & -\sin\left(\omega t + \frac{2}{3}\pi\right) \end{bmatrix} \cdot \begin{bmatrix} X_A \\ X_B \\ X_C \end{bmatrix}
\tag{4}
$$

Based on the above mathematical relations, the ATRU converter output voltage can be controlled by a combination of eight switching states. These states are described by using vectors *V (0)* to *V (7)*. Vector positioning states in the *d-q* reference system depend on the items in the control state in the *d-q* system, as illustrated in the figure below (Figure 5). It should be noted that for practical applications, six states are taken into account, defining the ATRU system operating voltage and ensuring appropriate values of the output voltage.

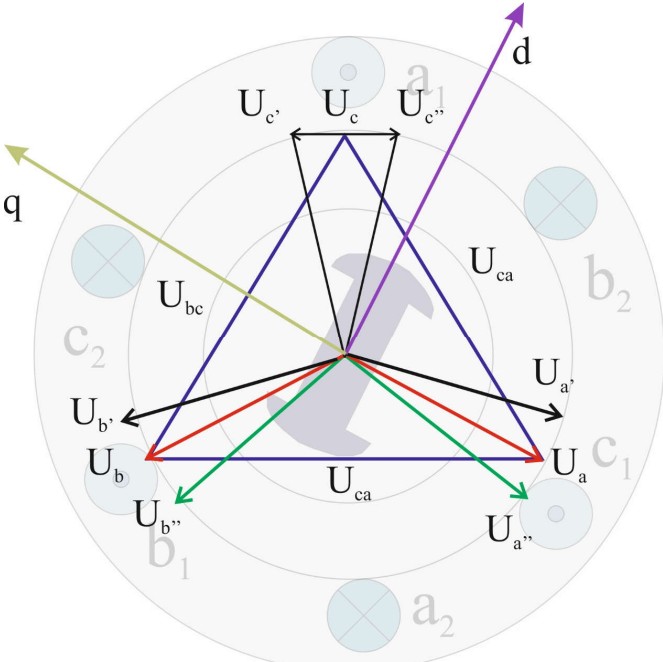

**Figure 5.** Voltage vector relations in the *d-q* reference system.



Where: $U_a$, $U_b$ i $U_c$- three-phase voltages occurring in the electrical machine PMSM, $U_{a'}$, $U_{b'}$ i $U_{c'}$- three-phase voltages in phase a, b and c shifted by angle $-\frac{2}{3}\pi$, $U_{a''}$, $U_{b''}$ i $U_{c''}$- three-phase voltages in phase a, b and c shifted by angle $+\frac{2}{3}\pi$, $U_{ab}$- three-phase voltage between phase a and b, $U_{ca}$- three-phase voltage between phase c and a, $U_{bc}$- three-phase voltage between phase b and c.

In accordance with the switching diagram shown in the figure above, it is possible to switch the transformation of three-phase AC voltage and current into DC by means of eight methods. This means that the system uses six switching states along with an additional two calculated from the switching period (T1, T2 and T0).

## 4. Exemplary Computer Simulations of the Selected AAES Power System Component in the Field of the PES System on the Example of the ATRU Unit in Accordance with the Concept of MEA

Simulation studies were carried out in the Matlab/Simulink programming environment, on the basis of which a mathematical model of the system was developed, with the following simplifying assumptions:

- the influence of magnetic hysteresis was omitted,
- the influence of temperature on the circuit parameters was omitted,
- ideal valves are adopted,
- the system is powered from a symmetrical three-phase source without a cable zero,
- the system is loaded with a current source.

The simplifying assumptions made cause minimal noise in the analyzed system of the electrical system of the aircraft, which translates into a better display of results and the appearance of additional higher harmonics in the AC voltage signal at the input of the ATRU unit. In addition, these assumptions translate into a smaller occurrence of eddy currents in the supply system of the ATRU unit, and thus to lower losses during idle operation. The mentioned assumptions also affect the correct cooperation of rectifier circuits with powered devices in the ATRU module, because they require them to generate electricity with certain parameters.

The most important parameters are voltage and current. Failure to observe the minimum permissible values of these parameters results in a malfunction or shutdown of the control system, and thus the generation of insufficient voltage and output current on the ATRU unit. Analogously, exceeding the nominal parameters may cause damage to the electronic components in the controllers.

During implementation of the ATRU unit control system, an algorithm is used, using the PWM signal. The basic property of such a follow-up control in a closed system is the induction of such a value of the current that is set by the preset signal. The sequence of pulses from the comparator is first synchronized by means of the D-trigger, synchronized by the generator of the synchronization pulse sequence, and then sent to the rectifier system as control pulses [20].

Analyzing the above-mentioned waveforms, one can conclude that the rectified output voltage builds up without over-voltages, and its determined value is established after six periods of the supply voltage. The value of the variable component in the rectified output voltage is determined to be below 3%. The applied control algorithm of the converter actually shapes the sinusoidal current in the reactor circuit.

In the figures above (Figures 6 and 7), the current, choke and set signal $U_z$ waveforms are presented in the form of rectified half-wave sine waves, while Figures 8 and 9 show the current and voltage waveforms of the ATRU unit. Moreover, it should be noted that the current of the choke repeats the shape of the set signal, where the instantaneous values of the choke current pulsate around the signal set in the zone as determined by the comparator hysteresis loop. In turn, the current shape approaches the signal given the smaller the value of the hysteresis zone. At the same time that the value of the hysteresis zone decreases, the carrying frequency of the transistor $T$ switches increases.

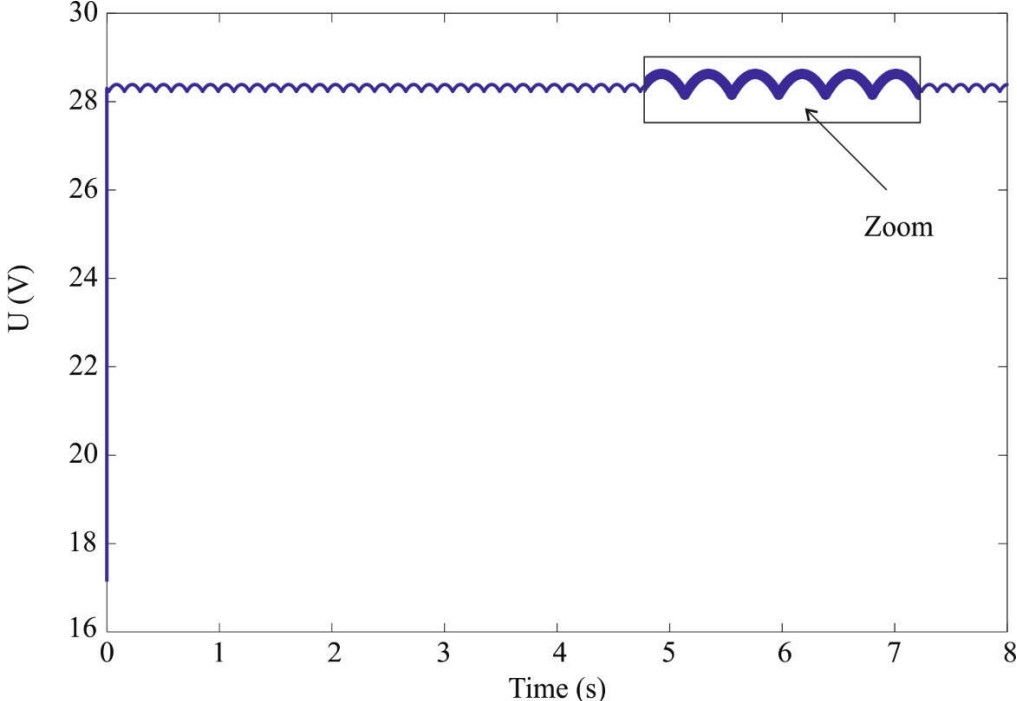

**Figure 6.** Voltage waveforms in the ATRU unit, based on the structure of a 12-pulse rectifier for output voltage 28 V.

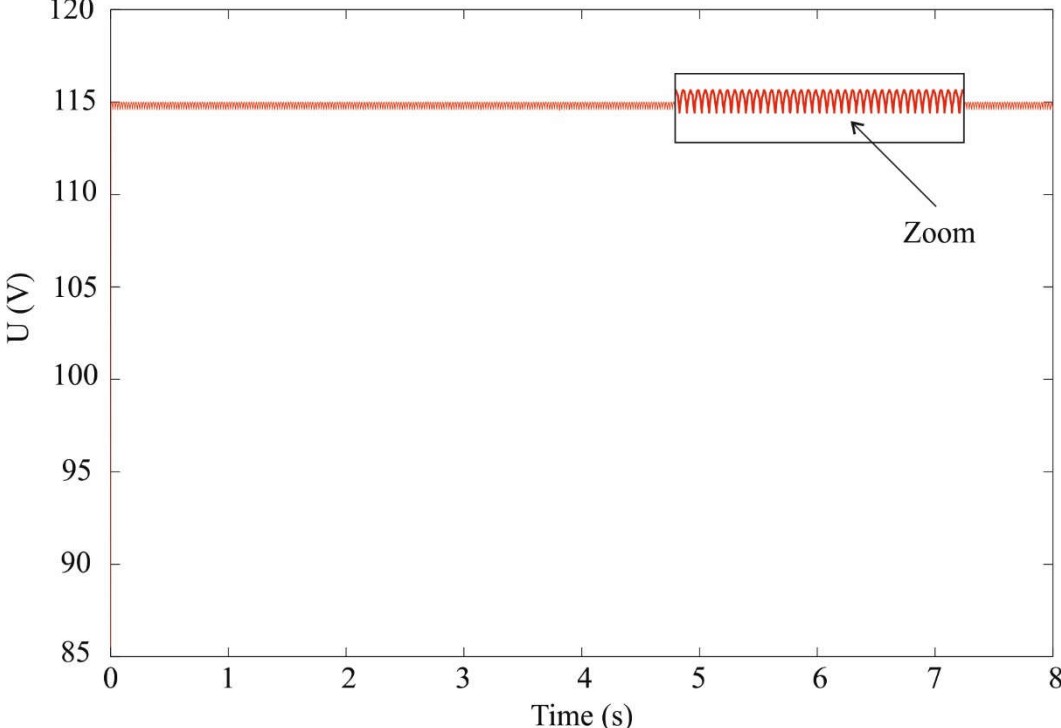

**Figure 7.** Voltage waveforms in the ATRU unit, based on the structure of a 12-pulse rectifier for output voltage 115 V.

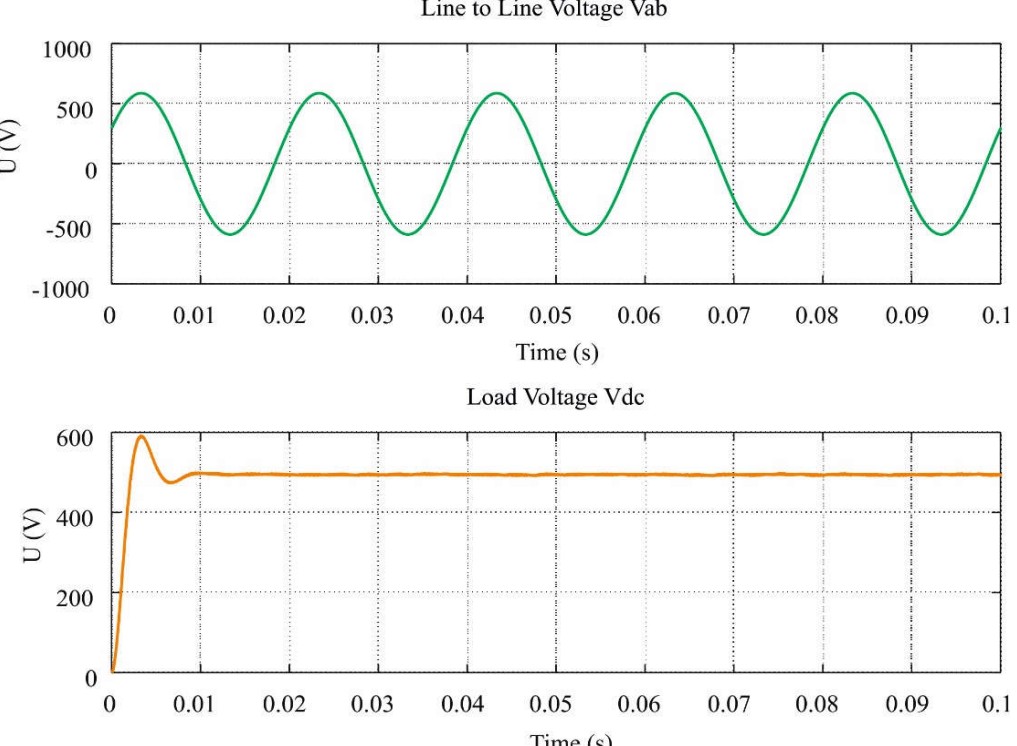

**Figure 8.** Voltage waveforms in the ATRU unit, based on the structure of a 24-pulse rectifier for output voltage 450 (V).

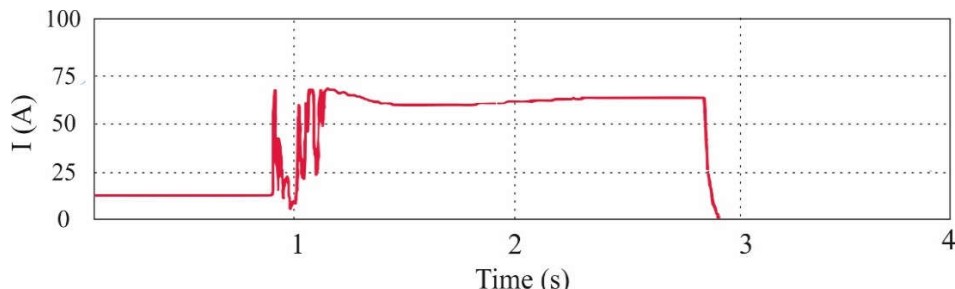

**Figure 9.** Current waveforms in the ATRU unit, based on the structure of a 24-pulse rectifier.

However, the maximum frequency of transistor switching is limited, both by the switching times of the force switches as well as by the power losses generated on the transistor associated with their switching.

Therefore, changes in the hysteresis zone width should be made taking into account the above parameters. On the basis of the measurements of voltages and currents, after analyzing the harmonic content of these waveforms and after determining the values of the transformer power reduction coefficients, it was found that in typical converter drive systems supplied from generators, the effect of strain on the supply voltage is ignored. Taking into account the above, only the degree of deformation of the load current is significant, i.e. the presence of higher current harmonics causing additional heating of the ATRU unit through eddy current losses.

Currently, at the stage of the process of converting ATRU converter systems, the manufacturers take into account the influence of higher harmonics in terms of the increase of load losses. One of the methods of minimizing additional load losses caused by eddy currents is the use of interlacing within the parallel coil wires and the inter-coil connections. Additionally, regardless of the ATRU system implementation, it should be remembered that in the real converter drive system, the power transfer

capacity of the transformer at non-sinusoidal power consumption decreases in the range of about 15% in relation to the nominal current indicated on the sinusoidal load sine nameplate.

## 5. Summary and Conclusions

This article discusses multi-pulse rectifier circuits with a multi-wave network current waveform and a modulator in the DC circuit. The main system in which the simulations were carried out the ATRU unit, which is responsible for the electrical network of the aircraft, in line with the trend of a more electric aircraft (MEA) for "rectifying" the AC voltage.

The simulation tests carried out showed that increasing the number of steps in the output voltage means that the 12-pulse systems exhibit properties of 24- and 36-pulse circuits, which is characterized by a lower impact of the rectifier on the supply network. In addition, the simulated system tests performed with the additional winding of the connecting choke showed a significant improvement in the shape of the mains current, bringing its course to a sinusoidal level of 1.03%, which should be considered a very good result. It is also important, from the point of view of energy efficiency, to consider the low power of the modulating current source, which in this case is only 2.35% of DC power.

Additionally, based on the calculated parameters, it can be concluded that the inverter input current is practically compatible in phase with the supply voltage; therefore, the considered ATRU rectifier system does not take reactive power from the power source. In addition, the content of higher harmonics in the waveform of the input current causes deformation at the input terminals of the power converter and a resulting reduction in the converter power factor. However, both weakness as well as reduction of higher harmonics in the input current, due to the high carrying frequency of modulation, do not pose major difficulties. In summary, the creation of a simulation model of the PWM type converter in the Matlab/Simulink programming environment enabled the analysis of basic properties and the determination of energy indicators.

**Author Contributions:** Formal analysis, R.K.; Methodology, L.S.

**Funding:** This research received no external funding.

**Conflicts of Interest:** The authors declare no conflict of interest.

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
