# Peer review of "Examination of Multi-Pulse Rectifiers of PES Systems Used on Airplanes Compliant with the Concept of Electrified Aircraft"

_applsci, doi:10.3390/app9081520_

Round 1
Reviewer 1 Report
The authors present an interesting study on the role played by multi-pulse rectifier circuits on power electronic systems of military and civil aircrafts. With regard to their paper I would like the following observations:
The abstract section should follow the structure suggested by this Journal, i.e. it is important to first place this study in a broad context before highlighting the purpose of the study. The whole abstract should be an objective representation of the article.
Furthermore, I miss a clearer explanation of the applicability and importance of this study throughout this paper.
In order to improve the readability of this paper, abbreviations should be properly defined the first time they appear. The authors have not always followed this rule, e.g. “PES” in line 15.
In introduction section, the authors should mention the main aim of the paper and highlight the main conclusions.
Some typos should be revised. E.g. in lines 74-75 the authors have mixed bold and plain text in the title of section 2. But special care must be taken in the improper use of punctuation marks like hyphens, e.g. in line 62 (among many others).
In addition to what I mentioned above, in line 152 the authors should clarify the meaning of uSA(t), uSB(t) and uSC(t) in Equation 1.
Furthermore, the authors should follow a standard format for equation notation. Thus, they should always include the independent variable whenever they mention a function, e.g. in equation 2 they should use uSA(t), uSB(t) and uSC(t) instead of uSA, uSB and uSC.
Moreover, the authors should clarify the meaning of uDC and whether UDC represents a different function or not; in addition to explain the meaning of ILOAD.
Figure 6 and 7 show an improper last XTick in Matlab plot (over 0.04 seconds).
The authors should follow the same format for XLabels of figure 8 “Time (seconds)” instead of “T [s]”.
The same XLabel should be also used in Figure 9. Moreover, the authors should use the same font size as in previous figures.
The authors should clarify how the simplifying assumptions described in lines 184-188 can limit the obtained results and conclusions.
Author Response
Dear Professor,
Thank you very much for your valuable and accurate comments and insights in the scope of the review of our article with ID number: applsci-485188. All corrections have been made in accordance with your recommendation, namely:
1. We have corrected a part of the article abstract according to the structure recommended by this Journal in line 14-28.
2. We have added in the article a clearer explanation of the applicability and meaning of our study: partly in the summary, partly in the part of the introduction in lines 78-88 and partly in relation to the simplifying assumptions made in lines 232-242.
3. The paper defines abbreviations when they appear for the first time in order to improve the readability of this article and draws attention to the spelling in English.
4. All typos and the misuse of punctuation marks have also been corrected.
5. The article explains all the necessary markings in lines 190-192 and 195-199, the standard format of equations (equations 1-2) has been used.
6. The simulation program Simulink corrected the simulation time and modified the drawings (Fig. 6-9), paying particular attention to their format, font size and label.
7. It was explained how the simplifying assumptions made with regard to the results obtained were explained and the conclusions drawn in lines 232-242.
Yours faithfully,
Dr. Lucjan Setlak and Dr. Rafał Kowalik
Polish Air Force University

Reviewer 2 Report
(1) The authors have studied the role of rectifier in the power electronics system. The authors have used Matlab program to conduct their analysis. The manuscript is technically sound and well discussed.
(2) For the thryristors order configuration, is there any reason why it has to be T3-T1-T2-T2-T1-T3-T3-T1? Please elaborate on this choice. What will happen if you use different order?
(3) The authors need to check their language use in the manuscript. Terms such as "you" should not be used in a professional journal paper.
(4) The manuscript can be accepted after minor revision on the English usage.
Author Response
Dear Professor,
Thank you very much for your valuable and accurate comments and insights in the scope of the review of our article with ID number: applsci-485188. All corrections have been made in accordance with your recommendation, namely:
1. We explained the order of inclusion of thyristors in line 152-161.
2. We paid detailed attention and made a correction in our article with regard to the proper use of English, among others with regard to not using the terms "you" in line 33-37 in a Scientific Journal.
Yours faithfully,
Dr. Lucjan Setlak and Dr. Rafał Kowalik
Polish Air Force University
